

# Genome-wide analysis of cellulose synthase (CesA) and cellulose synthase-like (Csl) proteins in *Cannabis sativa* L

Hulya Sipahi[1], Samuel Haiden[2] and Gerald Berkowitz[2]

[1] Department of Agricultural Biotechnology, Faculty of Agriculture, University of Eskişehir Osmangazi, Eskişehir, Türkiye
[2] Agricultural Biotechnology Laboratory, Department of Plant Science and Landscape Architecture, University of Connecticut, Storrs, CT, United States of America

## ABSTRACT

The cellulose and hemicellulose components of plant cell walls are synthesized by the cellulose synthase (CESA) and cellulose synthase-like (CSL) gene families and regulated in response to growth, development, and environmental stimuli. In this study, a total of 29 CESA/CSL family members were identified in *Cannabis sativa* and were grouped into seven subfamilies (CESA, CSLA, CSLB, CSLC, CSLD, CSLE and CSLG) according to phylogenetic relationships. The CESA/CESA proteins of *C. sativa* were closely related phylogenetically to the members of the subfamily of other species. The CESA/CSL subfamily members of *C. sativa* have unique gene structures. In addition, the expressions of four *CESA* and 10 *CsCSL* genes in flower, leaf, root, and stem organs of cannabis were detected using RT-qPCR. The results showed that CESA and CSL genes are expressed at varying levels in several organs. This detailed knowledge of the structural, evolutionary, and functional properties of cannabis *CESA/CSL* genes will provide a basis for designing advanced experiments for genetic manipulation of cell wall biogenesis to improve bast fibers and biofuel production.

## INTRODUCTION

*Cannabis sativa* L., an annual herbaceous plant, has versatile usage features as raw material in paper, textile, biofuel, automotive, and construction industries, mainly due to its cell wall content and other strains are useful as food and pharmaceuticals. Hemp is a preferable fiber source to cotton and other petroleum-derived synthetic fibers because it can be grown even in areas where water, fertilizer, and pesticide use is limited, and it yields a large amount of biomass in a short time (*Manaia, Manaia & Rodriges, 2019*). The hemp bast fiber, the outer part of the hemp stem known as phloem fiber, consists of cellulose microfibrils, which are contained in a matrix of hemicellulose and lignin (*Behr et al., 2016*).

The production and processing of hemp fiber in textiles depend on the amount and distribution of cellulose, hemicellulose, and lignin components in the fiber, which affects the mechanical properties of the fiber. Hemp fibers consist of 53–91% cellulose, 4–18%

Corresponding author
Hulya Sipahi,
hulya.sipahi@ogu.edu.tr,
hulyasipahi5@gmail.com

hemicellulose, 1–17% pectin, and 1–21% lignin, depending on the growing conditions, harvest year and location (*Liu et al., 2017*).

Cellulose is synthesized in the plasma membrane and located in the cell wall as microfibrils, which are the long chain structure of 1,4-D-glucose units connected by glycosidic bonds (*Mujtaba et al., 2017*). These microfibrils constitute the scaffold of the cell wall. The stiffness and organization of cellulose microfibrils in the cell wall affect cell growth and increase the cell's resistance to osmotic pressure (*Cosgrove, 2005*; *Hu et al., 2018*). The second major structural component of the cell wall is hemicellulose. Hemicellulose, which has branched polymers consisting of 500–3000 sugar units, is a heteropolymer composed of different types of hemicellulose such as xylan, glucuronoxylan, glucomannan, and xyloglucan. After cellulose fibers are synthesized in the cell wall, they are cross-linked with pectin and hemicelluloses.

Exposing the biosynthesis, assembly, organization, and networking of the cell wall load-bearing cellulosic fibrils is complex and essential (*Zhang et al., 2021*). Cellulose and hemicellulose polysaccharides are synthesized by the enzymes of the cellulose synthase A (CESA) family and the cellulose synthase-like (CSL) family, respectively. These families belong to the *glycosyltransferase 2* (*GT2*) superfamily (*Richmond & Somerville, 2000*). In Arabidopsis, ten members of the cellulose synthase (CESA) family have been identified, designated as CESA1 to CESA10. Among these, CESA1, CESA3, and one member from the group consisting of CESA2, CESA5, CESA6, or CESA9, are assembled into the Cellulose Synthase Complex responsible for cellulose synthesis in the primary cell wall (*Zhang et al., 2021*). Another distinct Cellulose Synthase Complex, composed of CESA4, CESA7, and CESA8, is involved in cellulose synthesis in the secondary cell wall (*Taylor et al., 2003*).

The genes of the cellulose synthase-like (CSL) family, grouping in from *CSLA* to *CSLG*, encode enzymes that generate hemicellulose, including xylans, xyloglucans, mannans, glucomannans and $\beta$-(1,3;1,4) glucan (*Holland et al., 2000*). A total of 30 genes belonging to the Cellulose synthase-like (CSL) family have been identified in Arabidopsis. *CSLA, CSLC,* and *CSLF* are responsible for the biosynthesis of mannan, xyloglucan, and (1 → 3; 1 → 4)- $\beta$-D-glucan, respectively (*Arioli et al., 1998*; *Richmond & Somerville, 2000*; *Richmond & Somerville, 2001*; *Lerouxel et al., 2006*; *Cocuron et al., 2007*; *Dwivany et al., 2009*; *Doblin, Pettolino & Bacic, 2010*). *CSLD* genes are involved in synthesizing xylan, homogalacturonan, and mannan (*Bernal et al., 2007*). *CSLJ* are responsible for (1, 3; 1, 4)-$\beta$-glucan biosynthesis (*Little et al., 2018*).

Because *CESA* and *CSL* genes play a crucial role in plant growth, immunity responses to pathogens, and plant biomass increase, genome-wide characterization of cellulose synthase and cellulose synthase-like gene families have been studied in a variety of plants, such as rice (*Hazen, Scott-Craig & Walton, 2002*), flax (*Guo et al., 2022*), pineapple (*Cao et al., 2019*), maize (*Appenzeller et al., 2004*), tomato (*Song et al., 2019*), tea (*Li et al., 2022*) and strawberry (*Huang et al., 2022*). Subsequently, over-expression and silencing experiments of certain *CESA* and *CSL* genes were performed to identify their roles in plant development and growth, plant phenotype and defense, cell wall integrity, and osmotic stress (*Held et al., 2008*; *Zhu et al., 2010*; *Chowdhury et al., 2016*; *Douchkov et al., 2016*; *Mabuchi et al., 2016*; *Mazarei et al., 2018*; *Huang et al., 2022*; *Li et al., 2022*; *Zhao et al., 2022*). For example, rice
*OsCSLD4* may regulate the function of cell wall localized proteins through its effect on cell wall polysaccharide composition, which in turn modulates intracellular signaling and regulates ABA synthesis, which has a critical role in regulating plant responses to salt stress, growth and development (*Zhao et al., 2022*). In a study on the role of CSLs in plant defence, down-regulation of glucan synthase-like 6 gene (*HvGsl6*) in barley led to lower papillary and wound callose deposition and increased cell wall penetration of *Blumeria graminis* f (Bgf) (*Chowdhury et al., 2016*). Thus, it has been proven that the *HvGsl6* plays a role in the deposition of callose, which contributes positively to the penetration resistance mechanism of Bgf. Likewise, the papillae of *HvCslD2*-silenced barley were more successfully penetrated by host-adapted, virulent, and avirulent non-host isolates of the powdery mildew fungus, and this papillary penetration was associated with lower cellulose content in the epidermal cell walls and increased digestion by fungal cell wall-degrading enzymes (*Douchkov et al., 2016*). Thus, barley CslDs have been identified to exhibit functional diversity, such as cell wall biosynthesis in dividing cells or local cell wall reinforcement during pathogen invasion attempts (*Douchkov et al., 2016*).

Although cellulose/hemicellulose is important in increasing the economic value of cannabis in terms of protection from environmental effects such as disease resistance and providing raw materials for industrial applications such as textiles and biodiesel, there is no report on the genome-wide identification and functional analysis of cellulose/hemicellulose synthesis genes in cannabis. Therefore, this study aimed to identify *CESA/CSL* gene family members in the cannabis genome and reveal their structure and evolution relationships. In this study, detailed identification of *CESA/CSL* genes and analysis of their gene expression in tissues will create a foundation for understanding their structural, functional, and evolutionary properties.

## MATERIALS & METHODS

### Identification of *CESA/CSL* gene family in *C. sativa*

The cellulose synthase (AtCESAs) and cellulose synthase like (AtCSLAs, AtCSLBs, AtCSLCs, AtCSLD, AtCSLE, AtCSLG) protein and coding sequences retrieved from TAIR (https://www.arabidopsis.org/) were used to determine *CESA/CSL* genes in three *C. sativa* genomes (NCBI Genome assembly ASM2916894v1, GenBank: GCA_029168945.1 Pink pepper (cultivar); Genome assembly JL_Mother, GenBank: GCA_012923435.1, Jamaican Lion ♀, isolate mother; Genome assembly JL_Father, GenBank: GCA_013030025.1, Jamaican Lion ♀ isolate father) using BlastP (*E*-value of $1e^{-5}$) and TBLASTN in NCBI. The Arabidopsis loci used as query are listed Table S1. After blast analysis, the candidate CESA/CSL proteins were validated by checking Pfam domains of the cellulose synthase and cellulose synthase like using HMMER v2.43 online program using default parameters (https://www.ebi.ac.uk/Tools/hmmer/). CESA family members were recognized by containing RING/U-box type zinc-binding domain (PF14569) and cellulose synthase (PF03552) domain. Those with glycosyltransferase-like family 2 (PFAM 13641) and glycosyltransferase family group 2 (PF13632) domains were characterized as CSLA and CSLC genes, respectively. Cellulose synthase-like D proteins were distinguished by

containing both RING/Ubox-like zinc-binding (PF14570) and cellulose synthase (PF03552) domains. All the other cannabis cellulose synthase-like proteins contained the cellulose synthase domains (PF03552) and they were classified according to phylogenetic similarity to CSLB, CSLC, CSLD, CSLE, CSLF, CSLG, CSLH, CSLJ and CSLM family members obtained from *A. thaliana*, *Oryza sativa*, *Linum usitatissimum*, *Sorghum bicolar*, *Solanum lycopersicum*, *Zea mays*, *Glycine max*, *Setaria italica* genomes (Table S1). The identified cellulose synthase and cellulose synthase-like genes in three different cannabis genomes are listed in Tables S2 and S3. Further bioinformatics analyzes were performed for genes of cannabis genome ASM2916894v1.

## Phylogenetic analysis

Amino acid sequences of CESA and CSL family members from *A. thaliana, C. sativa*, *Oryza sativa* and *Linum usitatissimum,* CSLMs from *Solanum lycopersicum* and *Glycine max*, and CSLJs from *Sorghum bicolar, Zea mays* and *Setaria italica* were used to construct phylogenetic tree. ClustalW was used to align CESA/CSL protein sequences. The evolutionary distances were calculated by the p-distance method (*Nei & Kumar, 2000*). The phylogenetic tree was constructed using maximum likelihood method with 1000 bootstrap replicates in MEGA11 (*Tamura, Stecher & Kumar, 2021*).

## Gene structure, motif identification, subcellular prediction, and chromosome localization

Gene Structure Display Server v2.0 (https://gsds.gao-lab.org/) was used to analyze the exon-intron structure of these genes (*Hu et al., 2015*). The MEME program (http://meme.sdsc.edu/meme/cgi-bin/meme.cgi) analysed the protein sequences to detect the motifs with the motifs number set 15, and other options were default (*Bailey et al., 2009*). Protein subcellular localizations were predicted by WoLF PSORT with plant parameters. (https://wolfpsort.hgc.jp/). According to WoLF PSORT results, HeatMap was constructed using TBtools software (*Chen et al., 2020*). The chromosome distribution of all *CESA/CSL* genes of cannabis ASM2916894v1 genome was visualized with TBtools software (*Chen et al., 2020*).

## Cis-element analysis of putative promoter regions, Ka/Ks calculation, and synteny analysis

The 2 Kbp upstream regulatory regions upstream from the start site of translation of *CESA/CSL* genes were retrieved from the NCBI website (https://www.ncbi.nlm.nih.gov/). The PlantCARE online software (http://bioinformatics.psb.ugent.be/webtools/plantcare/html/) (*Lescot et al., 2002*) was used to investigate the putative *cis*-regulatory elements in these promoter region sequences. HeatMaps were drawn using TBtools software (*Chen et al., 2020*). Gene duplications were determined by considering the length of the aligned sequence as covering >80% of the longer gene and their similarity being >80%. The Ka/Ks ratios were calculated using TBtools software (*Chen et al., 2020*). Synteny analysis was carried out using genome files of between *C. sativa* and *A. thaliana.* Reference genome information of the species used in the synthesis analysis is given in Table S4. TBtools

software (*Chen et al., 2020*) with an e-value 1e−10 and five BLAST hit cutoffs was used for synteny analysis.

## Plant material

The hemp variety used in this study was 'Wife.' Cuttings of this variety were taken from 5-month-old 'Wife' female plants and aeroponically rooted in an EZ-Cloner Classic Chamber™ (Sacramento, CA, USA). Cuttings were treated with Hormodin powder and placed in rock wool cubes soaked in 20 mL/L Clonex Nutrient Solution (Growth Technologies Ltd., Taunton, UK). Cuttings were rooted for three weeks before planting. Rooted cuttings were potted in a 3-gallon container with Pro-Mix HP (PRO-MIX, Quakertown, PA, USA) and fed twice a week with Botanicare™ liquid fertilizer (Vancouver, WA, USA). At the end of 8 weeks, they were transplanted to a 10-gallon container. During vegetative growth, plants were grown in a greenhouse at 25 °C under a daily 18 h light/6 h dark cycle, providing liquid feed in Jack's Nutrients (Jr. Peters, Inc., Allentown, PA, USA) containing 100 ppm N at each irrigation. During flowering, plants were grown under a 12 h light/12 h dark cycle for seven weeks and irrigated with 15-30-15 (NPK) Jack's Nutrient (Jr. Peters, Inc., Allentown, PA, USA) at 100 ppm N.

## RNA isolation, cDNA and RT-qPCR gene expression analysis

The amount of 100 mg of plant tissues (root, stem, leaf, and flower) was collected and immediately frozen in liquid nitrogen. RNA was extracted from leaves using The NucleoSpin Plant and Fungi RNA Isolation Kit (Macherey-Nagel, Düren, Germany). cDNA was synthesized from 2 μg RNA using the iScript Reverse Transcriptase Master Mix (BioRad, Hercules, CA, USA). qPCR analysis was carried out using Bio-Rad CFX. iTaq Universal Sybr Green Master Mix (BioRad, Hercules, CA, USA) was used. Selected *CESA/CSL* gene-specific primers were designed by PerlPrimer software (v1.1.21) (PerlPrimer for MicrosoftWindows, Owen J. Marshall, Australia) and listed Table S5. The specificity of primers were assessed using NCBI primer blast (https://www.ncbi.nlm.nih.gov/tools/primer-blast/). CsUbiquitin (CsUBQ, NCBI GenBank: JP465573.1) was used as the internal reference (*Guo et al., 2018*). Primer pair efficiencies were estimated using a standard curve from a serial dilution of cDNA from roots. Four serial dilutions were made to determine the quantification cycle (Cq) and reaction efficiencies for all primer pairs. Standard curves were generated for each primer pair by plotting the Cq value against the logarithm of each cDNA dilution factor. The $2^{-\Delta\Delta CT}$ method (*Livak & Schmittgen, 2001*) was used for gene expression analysis. qPCR conditions included a hold time of 90 °C for 3 m, 39 cycles of 95 °C for 10 s, 53 °C for 30 s, and 72 °C for 10 s. qPCR experiments were conducted with three or four biological replications with two technical repeats.

## Statical analysis

While paired-t test was used to compare the means of 2 dependent groups, *F* test (repeated measure Anova) was used to compare the means of 3 and 4 dependent groups. In Anova, Bonferroni multiple comparison test was applied for statistically significant group averages to assess variations among different tissues at a 5% significance level. Results are presented

in lower case. While there is no significant difference between those marked with the same letter, there are significant differences between those marked with different letters.

## RESULTS

A total of 29, 30 and 39 *CESA/CSL* genes were identified in ASM2916894v1 (cultivar pink pepper), JL_Mother and JL_Father (cultivar Jamaican Lion ♂) genomes of *C. sativa*, respectively (Tables S2 and S3). Two conserved domains, namely RING/U-box type zinc-binding domain (PF14569) and cellulose synthase (PF03552), were identified in the cellulose synthase A protein (CESA). The structural characteristics of the cellulose synthase-like proteins showed that CSLA proteins (glucomannan 4-beta-mannosyltransferase 9) and CSLC (xyloglucan glycosyltransferase) contained glycosyltransferase-like family 2 (PFAM 13641) and glycosyl transferase family group 2 (PF13632) domains, respectively. CSLD (cellulose-synthase-like D1) proteins included RING/Ubox like zinc-binding (PF14570) and/or cellulose synthase (PF03552) domains. Finally, the CSLB, CSLE, CSLG cellulose synthase-like proteins contained the cellulose synthase domain (PF03552). Also, all proteins included transmembrane and signal domains.

A total of eight *CsCESA* and 21 *CsCSL* genes (3 *CsCSLA*, 2 *CsCSLB*, 4 *CsCSLC*, 5 *CsCSLD*, 5 *CsCSLE*, 2 *CsCSLG*) from ASM2916894v1 genome were numbered according to their chromosomal locations and named species names (Table S2), These 29 genes were analyzed bioinformatically.

Based on the phylogenetic classification, eight CsCESA proteins were grouped in the CESA clade, which is the largest clade. The cellulose synthase-like proteins were classified into six CsCSL subfamilies (CSLA, CSLB, CSLC, CSLD, CSLE, CSLG) (Fig. 1). Eexceptionally, while no any CsCSL proteins from the pink pepper genome were clustered in the CSLM subgroup, only one protein from the JL_Mother genome (Fig. S1) and two from the and JL_Father genome were clustered in this subgroup (Fig. S2).

Many cellulose synthases were predicted to be localized mainly in the plasma membrane and endoplasmic reticulum, followed by the nucleus, vacuole, mitochondria and Golgi body (Fig. 2). Only CsCSLB2 was present in the chloroplast and CsCESA1 and CSLA2 in the extracellular membrane.

All *CsCESA/CsCSL* genes detected in cannabis genome (ASM2916894v1) were mapped and unequally distributed on all chromosomes of *C. sativa* except chromosome 7 (Fig. 3). The highest number of genes were found on chromosome 1 with six, while the lowest number was on chromosome 2 and 9. Also, a total of two duplication events were observed.

Cis-acting elements are conserved nucleotide sequences in the gene's promoter to which transcription factors can bind, and they regulate the transcription of the gene of interest. Variations in the Cis-acting elements in the promoter regions of these genes may lead to variations in the phenotypic characteristics of the organism, such as its development and response to biotic and abiotic factors. A two kilo base upstream sequence of the *CsCESA/CsCSL* genes was searched to identify cis-acting elements. These cis-elements were categorized into four main groups: light-responsive elements, environmental stress-responsive elements, hormone-responsive elements, and development-related elements.

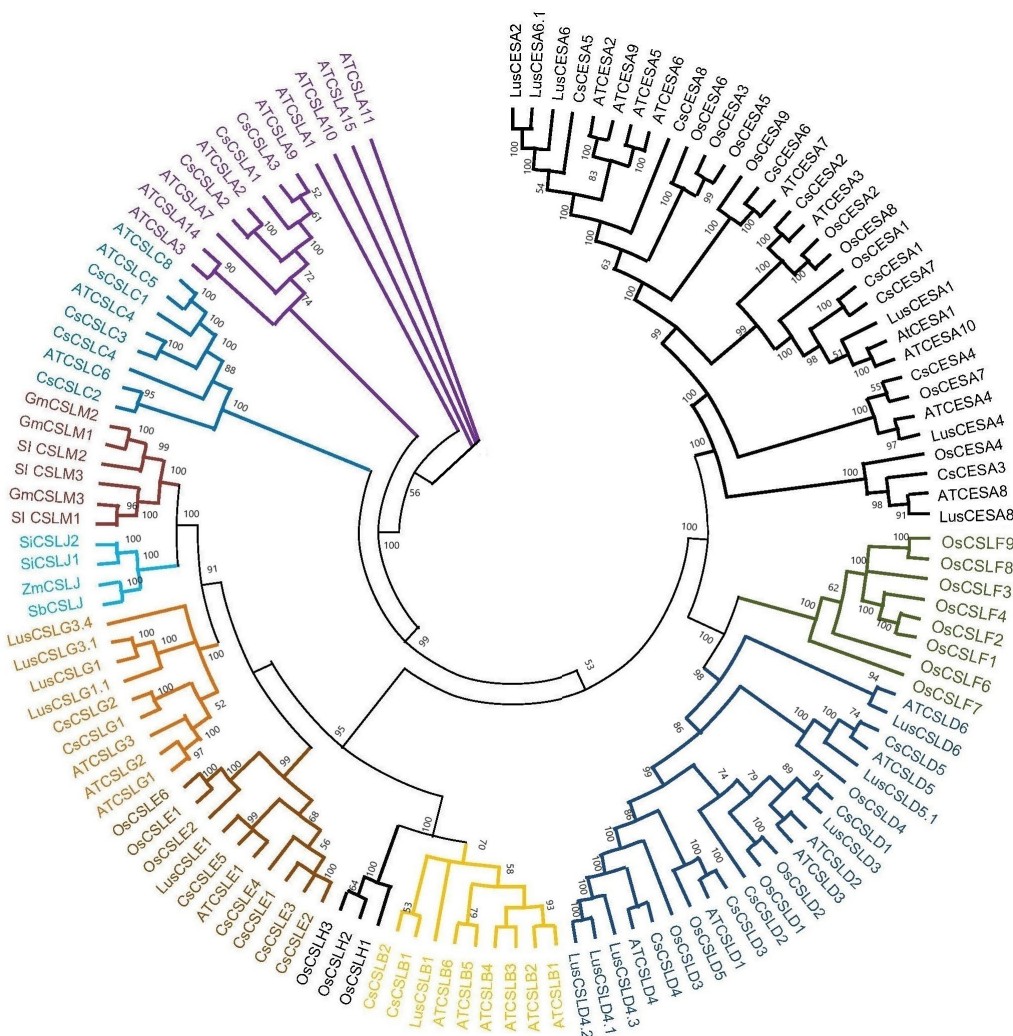

**Figure 1** **The neighbor-joining phylogenetic tree of *CsCESA/CsCSL* proteins from *Cannabis sativa* and *Arabidopsis thaliana*.** The evolutionary tree was conducted using MEGA11 with 1,000 bootstrap replicates. The tree was divided into seven groups: Group I, G.

Twenty-five light-responsive elements were found (Fig. 4A). All but one *CsCESA/CsCSL* contained the Box 4 motif, making it the most abundant light response element. Thirteen cis-acting elements were involved in environmental stress response (Fig. 4B). MYB (drought and ABA signaling), MYC (drought, salt, and stress response), STRE (multiple signaling including heat stress), LTR (low-temperature response), ARE (anaerobic induction) motifs were most abundant in *CsCESA/CsCSL* gene promotors. Thirteen cis-acting hormone response elements were detected (Fig. 4C). ABRE (abscisic-acid-responsive element), ERE (the ethylene-responsive element), CGTCA motifs and TGACG motifs (MeJA-responsive elements) were the most common. *CsCESA/CsCSL* genes carrying hormone response cis-regulatory elements might be upregulated by these hormone treatments. TGA-box (auxin-responsive element) was found only in one *CsCSLE4*. Finally, fourteen development

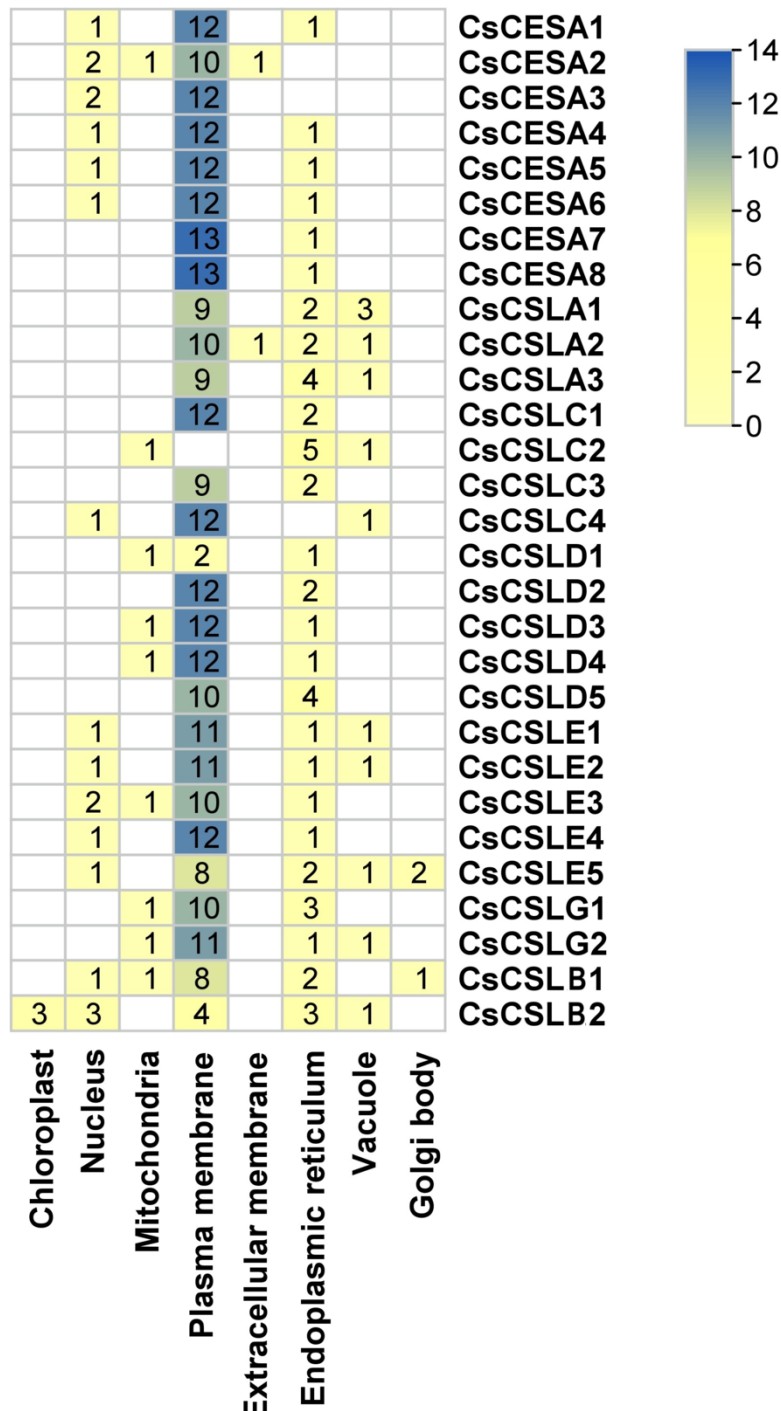

**Figure 2 Subcellular localization of *CsCESA/CsCSL*.** Plas, the plasma membrane; Nucl, nucleus; E.R., endoplasmic reticulum; Mito, mitochondria; Extr, ekstra cellular; Vacu, the vacuole; Cyto_plas, cytoplasm; Golg, golgi body; Chlo, chloroplast; Cyt, cytoplasma.

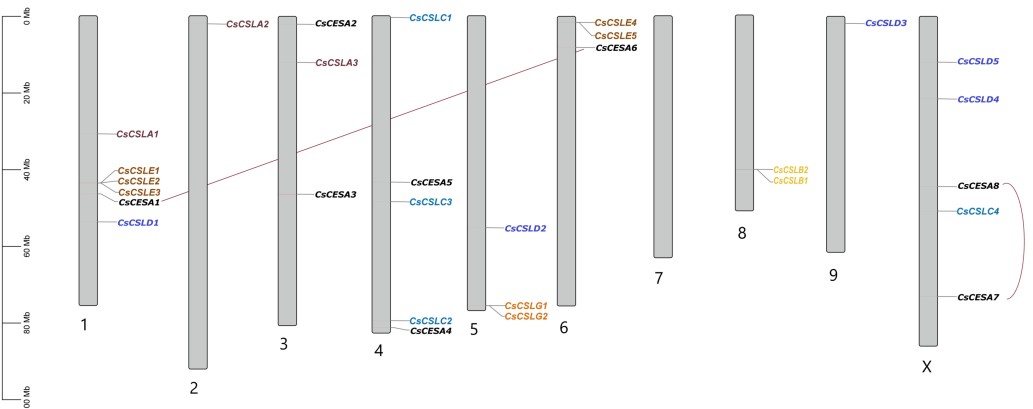

**Figure 3** Chromosomal distribution of 29 *CsCESA/CsCSL* genes in *Cannabis sativa*.

response cis-acting elements were found (Fig. 4D). Among them, the AAGAA-motif (the endosperm-specific negative expression), O2-site (zein metabolism regulation), and as-1 (the root-specific expression) were found to be the most abundant elements.

A total of 15 conserved protein motifs were searched (Fig. 5B). Consistent with conserved domain analysis, CsCSLA shared similar motif composition with CsCSLC, CsCSLG with CsCSLE and CsCSLB, and CsCESA with class CsCSLD. CsCESA and CSLDs have the highest number of motifs, with 15 motifs.

Considerable diversity of the exon-intron structures of the *CsCESA/CsCSL* genes was noted (Fig. 5C). *CsCESA* genes usually have 14 exons, with the exciting exceptions that *CsCESA1* has only one exon, while the longest gene, *CsCESA3* has 26 exons. Although the exon-intron numbers of the *CsCSLA* and *CsCSLB* genes were the same within the members of their subfamilies, diversities were observed in the lengths of the introns. *CsCSLC* and *CsCSLD* genes have between four to five exons. The highest intron length variation was in the *CsCSLE* subfamily, with eight exons.

Syntenic maps were constructed between *CESA/CSL* genes of *C. sativa* and *A. thaliana* (Fig. 6, Table S4). Nine *C. sativa CESA/CSL* genes were found orthologous to *A. thaliana* genes.

The nonsynonymous (Ka)/synonymous (Ks) ratios were calculated to reveal the evolutionary status of the two linked CsCESA/CsCSL gene pairs (Table S6). The Ka/Ks ratios of two linked *CsCESA* gene pairs (*CsCESA1/CsCESA6* and *CsCESA7/CsCESA8*) were found to be less than one, indicating the presence of purifying selection and, in this case, the maintenance of the number of members in this gene family.

The members of the cellulose synthase family exhibit tissue-specific expression (*Hamann et al., 2004*). To investigate the expression patterns of *CESA/CSL* genes in different tissues (flower, leaf, root, stem) of cannabis, the expression of 14 *CESA/CSL* genes, selected based on phylogenetic groups, gene structures, and cis-elements was evaluated by RT-qPCR (Fig. 7, Table S7). All examined *CsCESA* genes were expressed in four different tissues with variable levels, suggesting that all these *CESA* genes are necessary for primary or secondary cell wall formation. One of the *CsCESA* genes (Fig. 7A) was expressed relatively

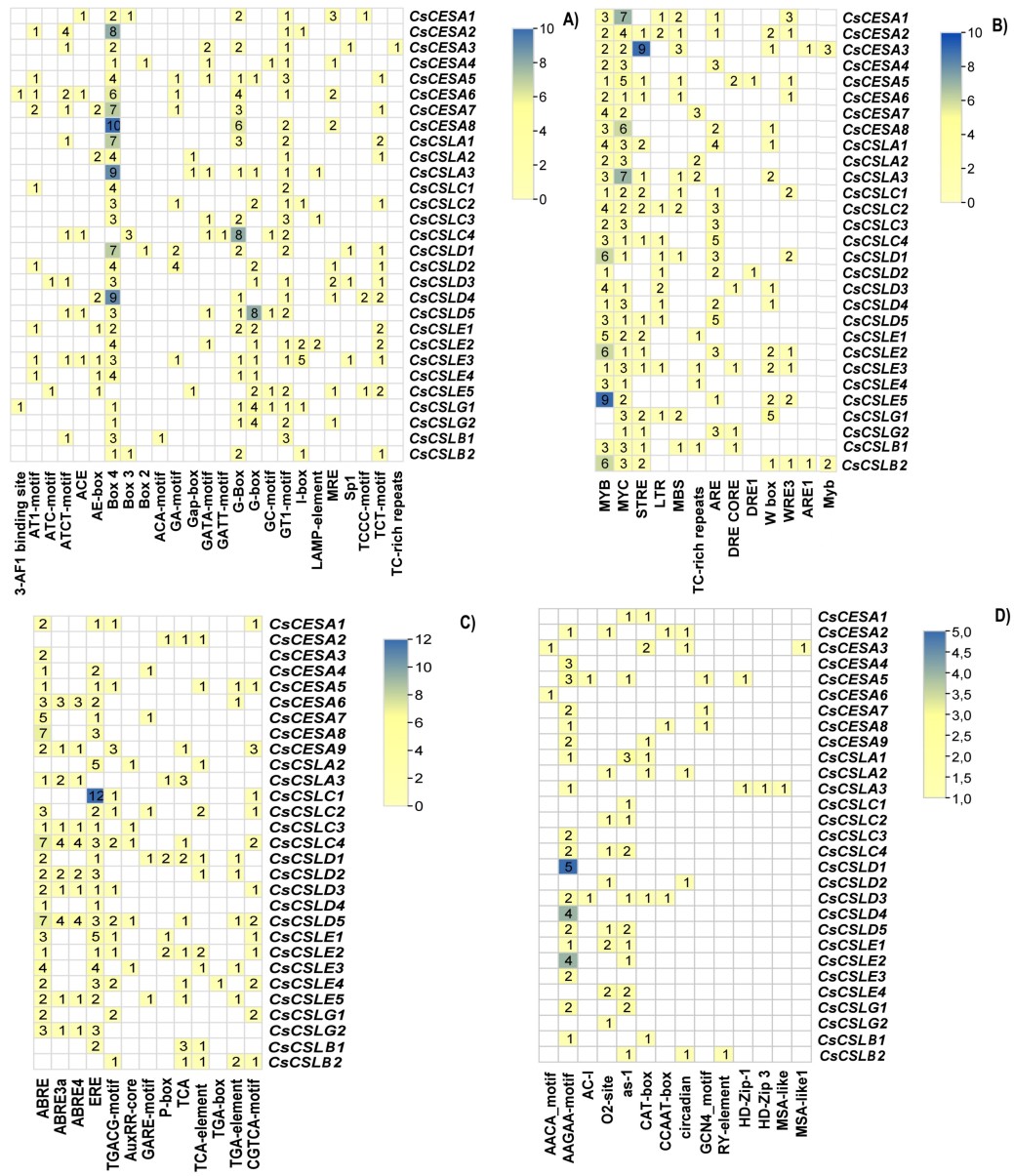

**Figure 4  Cis-acting elements found in *CsCESA/CsCSL* genes.** (A) Light responsive elements, (B) environmental stress response elements, (C) development response elements, (D) hormone response elements.

higher in flower and the other two (b, c) in leaf, but the difference in the expression levels of these genes between the tissues was not statistically significant. In addition, the expression level of one *CsCESA* gene (Fig. 7D) was highest in stem and lowest in flower. *CsCSLA* (Fig. 7F), *CsCSLB* (Fig. 7G), *CsCSLG* (Fig. 7M) showed good leaf-specific expression. *CsCSLC* (Fig. 7H) exhibited the highest expression in the flower, whereas *CsCSLD, CsCSLD, CsCSLG* genes (Figs. 7J, 7K and 7N) were expressed only in root and stem.

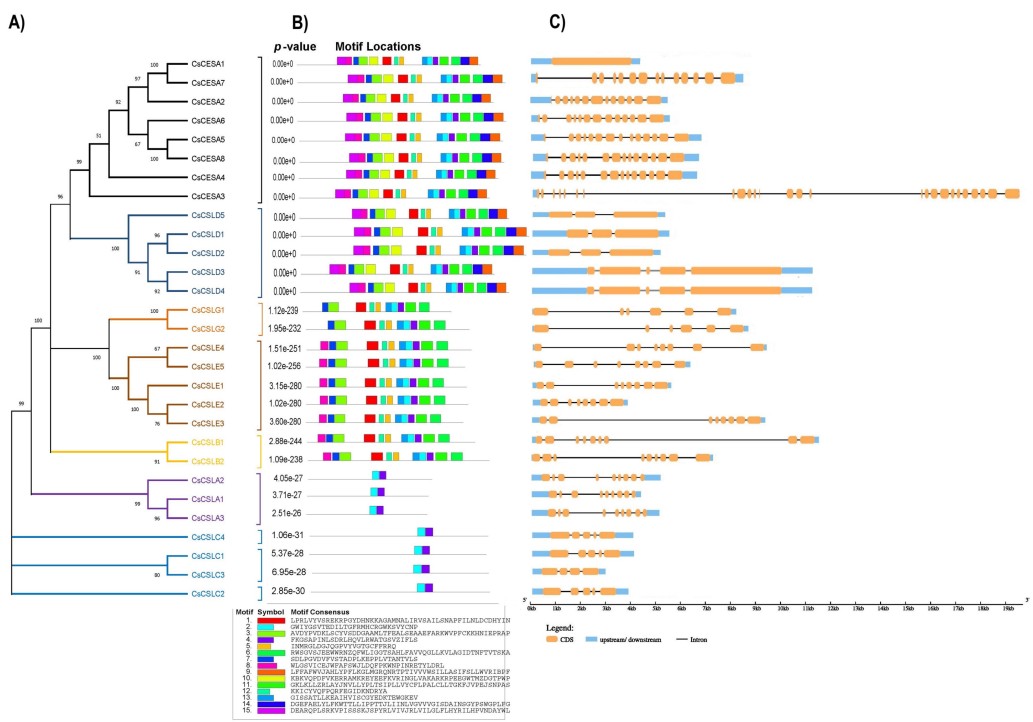

**Figure 5** (A) Phylogenetic relationships, (B) conserved motifs, (C) gene structures of *CsCESA/CsCSL*, the orange, blue, and black colors represent the exon CDS, UTR upstream/downstream, and introns, respectively.

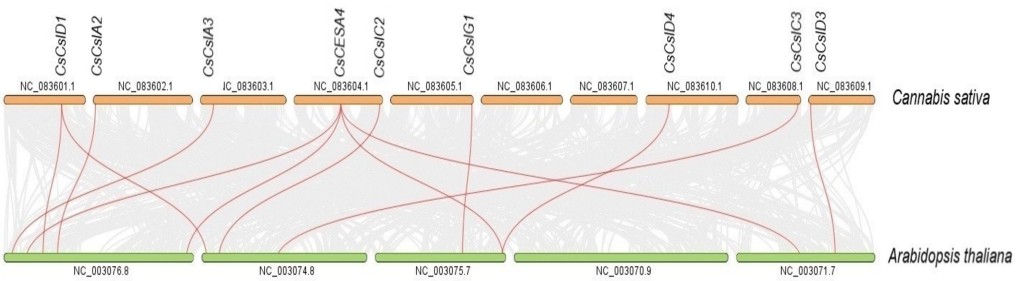

**Figure 6** Synthenic relationships between *Cannabis sativa*, and *Arabidopsis thaliana*. Red lines represent synthenic lines while grey color represents shared genomic blocks among them.

# DISCUSSION

The structure, composition, and organization of cell wall components have critical effects in terms of affecting the growth and development of the plant and its response to environmental stimuli by determining the cell's shape, strength, structural integrity and response to abiotic and biotic environmental stresses. Furthermore, the substances of the cell wall affect the quality and processing of the plant for use as paper, textiles, bioethanol, feed and food.

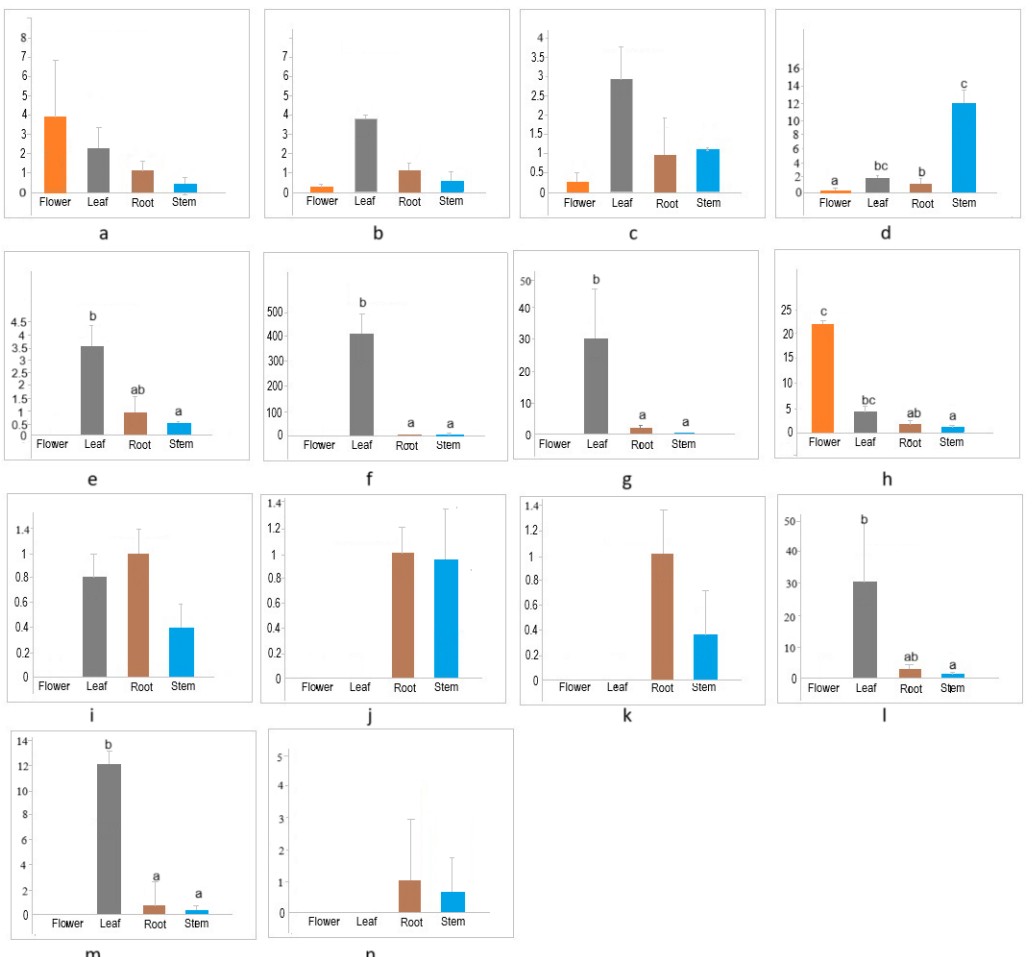

**Figure 7** Expression levels of *CsCESA/CsCSL* genes in different tissues.

The plant cell wall matrix mainly comprises polysaccharides such as cellulose, hemicellulose and pectin. The cellulose is embedded in the cell wall matrix as microfibrils. Microfibrils are a bundle of 40 cellulose molecules that run parallel to each other and are attached by hydrogen bonds. Each cellulose molecule is an unbranched polymer composed of glucose units linked by an $\beta$-1,4-bond. It is estimated that hundreds of enzymes are responsible for cell wall biosynthesis (*Keegstra & Raikhel, 2001*; *Scheible & Pauly, 2004*; *Liepman & Cavalier, 2012*). Cellulose Synthase (CESA) and Cellulose Synthase-Like (CSL) family proteins synthesize various $\beta$-glycan polymers. In this study, CESA and CSL family proteins were identified and characterized, and their phylogenetic relationships were revealed by using genome sequence and bioinformatics analysis tools in cannabis, which is a fiber and biofuel source and has a crucial phytochemical content.

To better understand the CsCESA/CsCSL characteristics in cannabis, genome-wide analyses were carried out *via* phylogenetic relationships with existing gene structures and expression in different tissues. 8 *CsCESA* and 21 *CsCSL* (3 *CsCSLA*, 2 *CsCSLB*, 4 *CsCSLC*, 5 *CsCSLD*, 5 *CsCSLE*, 2 *CsCSLG*) genes in cannabis ASM2916894v1 genome resemble the

CESA/CSL phylogenetic classification in other species. As expected, the monocot-specific CSLF, CSLH and CSLJ protein subfamilies were not found in the cannabis genome.

Syntenic relationships are shaped by conservation or DNA substitution rates among taxa and genes in the genomes of different species. Gene orthology relationship determined by synteny data analysis can support the phylogenetic relationships of multiple gene families (*Gabaldón & Koonin, 2013*). The orthologous relationships of Cs*CSL* genes with *A. thaliana*, with the exception of *CsCESA4* gene, supported their close relationship in the phylogenetic tree (Table S4, Fig. 1).

The existence of closely clustered CESA/CSL subfamily members between cannabis and other eight species used in phylogenetic tree resulted from evolutionary conservation and closer homology (Fig. 1). According to phylogenetic tree analysis (Fig. 1), CsCESA1/7 and CsCESA2 were closely clustered with AtCESA1 and AtCESA3, respectively. In addition, AtCESA5/6/2/9 are grouped with CsCESA5/8/6. AtCESA1, AtCESA3, and any of AtCESA2/ 5/ 6, or /9 complex cause primary wall cellulose accumulation with unequal enzymatic activity in cells undergoing cell division and elongation (*Persson et al., 2007*; *Hu et al., 2018*). The results inferred that these orthologous *CsCESA1/7/2*, or 5/8/6 genes may have similar functions.

The expression analysis showed that four *CsCESA* genes were expressed in four different tissues with varying levels. Only one *CESA* gene (Fig. 7D) were expressed at highest level in the stem. These results may support that *CsCESA* genes play roles in the growth of different tissues. In addition, Arabidopsis CESA4/7/8 are involved in secondary wall thickness by increasing the amount of cellulose (*Tanaka et al., 2003*; *Taylor et al., 2003*; *Zhong, Cui & Ye, 2019*) and phylogenetically clustered together with *CsCESA4/6/3* gene homologous, respectively (Fig. 1). These results suggest that *CsCESA4/6/3* might participate in establishing secondary cell walls.

Revealing gene structures and protein motif patterns of gene family members may elucidate the evolution and diversity of their structure and function. Most CsCESA and CsCSL subfamily members, closely grouped in the phylogenetic tree according to their amino acid sequences (Fig. 5A), share the same or similar motif distributions.

The motif components of CsCESA and CsCSLD were highly similar. However, the CsCSLA and CsCSLC genes had different motif patterns. Although the members of the CsCESA/CsCSL subfamily were generally similar in terms of exon and intron number and length, they also showed some differences (Fig. 5C). A similar situation has been noted for members of the DcCSLD subfamily in *Dendrobium catenatum*, and it has been reported that intron gain/loss may occur in the evolution of *DcCSLD* genes (*Xi et al., 2021*).

Several members of the CESA/CSL gene family have been reported to have varying expression levels depending on tissue type, growth stage, and environmental factors (*Cao et al., 2019*; *Li et al., 2020*; *Li et al., 2022*; *Song et al., 2019*; *Kaur et al., 2017*; *Yuan et al., 2021*; *Liu et al., 2022*; *Nawaz et al., 2017*; *Marcotuli et al., 2018*; *Hou et al., 2023*). Likewise, the results of this study identified different expression levels of several *CsCESA/CsCSL* genes in four tissues. *CSLA* (Fig. 7F), *CSLB* (Fig. 7G) and *CSLG* (Fig. 7M) were expressed mainly in the leaf. In contrast, *CsCSLD* (Figs. 7J and 7K) and *CsCSLG* (Fig. 7N) were expressed only in root and stem, whereas *CSLC* (Fig. 7H) was mainly expressed in the flower. In the

previous study conducted to determine transcriptomic changes associated with bast fiber development stage in textile hemp, certain CES/CSL family members showed differential expression patterns in the stem's upper, middle, and lower internodes (*Guerriero et al., 2017*). For example, the annotation against the *Arabidopsis* database showed that some contigs annotated with *CSLC5* were more highly expressed in the upper internode, with *CSLC04* in the middle internode, with *CSLE1, CSLG1* and *CSLB04* in the lower internode. The progressive decrease in expression from the top to the bottom of the stem was detected in *CSLC04* and *CSLC5* genes, while progressive increase in expression along the stem axis for *CSLE1, CSLG1, IRX1, CSLG3, CSLE1, CSLB04* genes.

In another important fiber plant, flax, expression differences were also found in different growth stages and stem parts, confirming the role of *CESA/CSL* genes in cell wall thickening (*Guo et al., 2022*). The flax *CESA3/8* (Lus10007538, Lus10007296) and *CSLD4* (Lus10008225) genes have been shown to be active during early fiber development. These genes are specifically expressed at the stage of fiber development when there is an increased amount of secondary cell wall deposition during the period of rapid growth. The flax *CESA6* genes (*Lus10006161.g* and *Lus10041063.g*) were found to be specifically expressed in the stem during fiber maturation. The expression levels of *LusCSLE1* (*Lus10016625*) and the two Lus*CSLG3* (*Lus10023056.g* and *Lus10023057.g*) genes were higher in 30 cm plants than in 50 cm plants. The opposite trend occurred for *CESA6 (Lus10006161.g* and *Lus10041063.g*) genes. The phylogenetic tree of flax and cannabis *CESA/CSL* genes showed that the flax *CESA/CSL* genes were closely clustered with those of cannabis (Fig. 1). This may indicate that flax and cannabis *CESA/CSL* genes may have similar functions during the fiber development stages.

In addition to improving fiber quality, a mutant allele of *CESA3* gene was developed using CRISPR/Cas9 base editing to improve herbicide resistance in Arabidopsis plants, and this mutant allele was found to confer plant resistance to the herbicide C17 (*Hu et al., 2019*). Shortly, editing *CESA/CSL* genes could lead to advances in areas such as enhancing plant growth and development, and improving disease and pest resistance. Also, the silencing or overexpression of genes associated with cell wall synthesis may indicate changes in plant-pathogen interactions in the cell wall, which may shed light on whether cannabinoid production is affected. In this study, the genome-wide identification of *CsCESA/CsCSL* genes will provide the basis for future investigation of their role.

## CONCLUSIONS

In this study, eight *CsCESA* and 21 *CsCSL* genes in *C. sativa* genome (ASM2916894v1) were identified according to their conserved domains and motifs. Their features were characterized, including gene structure, chromosome location, phylogenetic analysis and syntenic relationships. The cis-element analysis identified the multifunctional role of *CsCESA/CsCSL* genes in growth, hormone responsiveness, and biotic and abiotic stress responsiveness. *CsCESA/CsCSL* genes exhibited diverse expression patterns in flower, leaf, root, and stem tissues. The detailed characterization of CESA/CSL in cannabis may aid

in designing experiments for future genetic manipulation of cellulose and hemicellulose synthesis genes to breed cultivars with high fiber quality and bioethanol yield.

### Funding

This work was supported by funding by the Foundational Knowledge of Plant Products program at the USDA National Institute of Food and Agriculture (award 2022-67014-37049) and Hatch project CONS01027. There was no additional external funding received for this study. The funders had no role in study design, data collection and analysis, decision to publish, or preparation of the manuscript.

### Grant Disclosures

The following grant information was disclosed by the authors:
Foundational Knowledge of Plant Products program at the USDA National Institute of Food and Agriculture: 2022-67014-37049.
Hatch project: CONS01027.

### Competing Interests

The authors declare there are no competing interests.

### Author Contributions

- Hulya Sipahi conceived and designed the experiments, performed the experiments, analyzed the data, prepared figures and/or tables, authored or reviewed drafts of the article, and approved the final draft.
- Samuel Haiden performed the experiments, analyzed the data, prepared figures and/or tables, authored or reviewed drafts of the article, and approved the final draft.
- Gerald Berkowitz conceived and designed the experiments, authored or reviewed drafts of the article, and approved the final draft.

### Data Availability

 The raw data are available in the Supplemental Files.

### Supplemental Information

Supplemental information for this article can be found online at http://dx.doi.org/10.7717/peerj.17821#supplemental-information.

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
