# Peer review of "Genome-wide analysis of cellulose synthase (CesA) and cellulose synthase-like (Csl) proteins in Cannabis sativa L"

_PeerJ, doi:10.7717/peerj.17821_

## Round 0.1 · original submission · Major Revisions

Dear authors,

After careful consideration of the peer reviews for your manuscript "Genome-wide analysis of cellulose synthase (CesA) and cellulose synthase-like (Csl) proteins in Cannabis sativa L.," I recommend major revisions. I highlight the following issues in the manuscript:

- The manuscript needs better organization for clarity. A well-organized manuscript with distinct sections for introduction, methodology, results, discussion, and conclusion is crucial for readability and comprehension. This issue, pointed out by Reviewer 1, should be addressed in your revision.

- There are significant concerns about the experimental methods used, particularly the phylogenetic analysis. I agree with reviewer 3's feedback on the choice of methods and the absence of bootstrap support in phylogenetic analyses needs attention. Incorporating more robust phylogenetic methods and providing detailed statistical analyses will greatly enhance the scientific rigor of your study. It is crucial to address all points raised by reviewer 3. The inclusion of additional CesA/Csl sequences from other well-characterized plant groups, as suggested by Reviewer 2, is crucial. This would provide a broader context for your findings and enhance the comparative aspect of your study. I suggest adding 2 or 3 species, close to Cannabis and besides Arabidopsis in phylogenetic analyses.

The manuscript has several technical issues and typographical errors noted by the reviewers. These errors undermine the manuscript's credibility and must be thoroughly addressed.

I also strongly suggest incorporating proper statistical analysis and discussing primer efficiency in your qPCR data. This is essential for better analysis of your findings.

While all reviewer comments are valuable, the detailed feedback from Reviewer 3 should be a primary focus in your revisions. The insights into methodological improvements and comments of reviewer 3 are particularly critical for enhancing the quality of your manuscript.

**Language Note:** The review process has identified that the English language must be improved. PeerJ can provide language editing services - please contact us at [email protected] for pricing (be sure to provide your manuscript number and title). Alternatively, you should make your own arrangements to improve the language quality and provide details in your response letter. – PeerJ Staff

·

Basic reporting

The article entitled "Genome-wide analysis of cellulose synthase (CesA) and cellulose synthase-like (Csl) proteins in Cannabis sativa L." is not well-compiled manuscript, and the authors just performed bioinformatic analysis on the CesA gene family and conducted the expression verification in the different tissues. In general, the insufficient results are innovative, significant and useful for the research on Cannabis sativa, while the article structure was not clear enough.

Experimental design

The experimental design was scientific and rigorous, while insufficient for publishment.

Validity of the findings

The findings were ture and reliable.

Reviewer 2 ·

Basic reporting

no comment

Experimental design

no comment

Validity of the findings

The results and discussion need to be more clear and improvement. The discussion should be rephrased and organized.

Additional comments

Sipahi et al., analyzed cellulose synthase (CesA) and cellulose synthase-like (Csl) proteins in the Cannabis sativa L. The authors showed some interesting results about cannabis CesA/Csl genes evolution and functional properties. However, I have some comments about the analysis and results, see below.

1. The authors showed the topology of CesA/Csl only using the Arabidopsis, whether it’s enough to clearly identify all the proteins and show their relationships? I suggest the authors add more sequences from other well characterized CesA/Csl plant groups, such as rice, grass, etc. Furthermore, for the Figure 1, I suggest the authors label the bootstrap value and the CesA/Csl families.
2. In the part of expression levels of CsCesA/CsCsl genes in different tissues, some of the genes expressions have higher error bars, I suggest the authors manually check the results, and are the results strong enough to prove the author’s conclusions? In addition, I could not follow the results and discussion related to the expression levels of CsCesA/CsCsl genes. I suggest the authors rephrase these parts.
3. There are some typo errors throughout the manuscript, such as:
(1). Line 62, ‘synthasei A’.
(2). Line 69 and 75, ‘3-(1,3;1,4) glucan’, keep the font consistant.
(3). Line 136, ‘2g RNA’?
(4). Line 246, ‘Cellulose Synthase-L1ke’, like?

·

Basic reporting

no comment

Experimental design

The research described in this manuscript fits with the Aims and Scope of the journal. The research question is well defined and meaningful, but there could be more emphasis on how this research fills an identified knowledge gap, especially that a previous study is mentioned at some point.

Regarding the methods, a lot of details are lacking which would make the study difficult to replicate as it is. Moreover, some choices made by the authors make it difficult to consider this study rigorous: choice of neighbour-joining and absence of bootstrap support for the phylogenetic analyses, absence of statistical analyses for the RT-qPCR, confusion regarding the genome(s) that were used, confusion regarding the naming/grouping of the candidate genes identified.

L92: How were those genes identified in Arabidopsis? Was it based on Pfam or Interpro annotations, as described later for the candidates identified?

L92: See comment above for gene vs. protein naming nomenclature.

L94: Since there is a mention of e-value, I suppose BLAST was used, but this is not clear. Please clarify. Were the protein sequences used as query against the translated genomic sequence, or gene predictions? Which genome(s) of cannabis was(were) used? Table S1 suggest cs10, but this should be indicated.

L99-100: Please clarify « The redundant sequences were removed ». Identical sequences? Could they be real paralogues?

L103-109 (Phylogenetic analysis) : Neighbor-joining is a clustering algorithm that can make quick trees. This method is good to give you an idea about the data, but are not the most reliable. Other methods, such as maximum likelihood or Bayesian methods, can apply a model of sequence evolution and are ideal for building a phylogeny using sequence data. Moreover, while the authors claim to have used « 1000 bootstrap replicates », those are not presented on the phylogenetic tree (Figure 1), and it is thus hard to judge the validity of the analysis. It is also not clear how the naming was achieved. First, how was the numbering decided? Is it simply based on their genomic location (i.e. CesA1 is the first CesA appearing on chromosome 1, or the first chromosome harboring a CesA gene)? Then, how was it decided that CsCSLG2 and CsCSLG6 are part of the G family, and not another family absent in Arabidopsis (e.g. CSLM)? A similar problem is found for CsCSLE3 and CsCSLE8. Perhaps the authors should consider adding sequences from every clades (using other plant species). Similarly, how was decided that CsCSLH members are from that family, since there are no Arabidopsis counterparts?

L112-114: « to confirm the motifs ». Which motifs?

L114: Replace « locations » with « localizations ».

L114-115: Specify that WoLF PSORT was used with « plant » parameters. More importantly, the link provided is not valid. wolfpsort.hgc.jp appears to be the correct website.

L115: The authors should clarify the data that was used to produce the heat map (the results from WoLF PSORT).

L124: The provided link is not valid.

L125: Synteny analysis with rice is not mentioned, but is presented later.

L127-131 (Plant Material). Unclear. « maintained in this environment for 7 weeks ». Which environment? Vegetative growth? How much time in each stage? Jack’s Nutrients: please provide a source. Which one of their products was used during vegetative growth? How many plants? Were they clones? Females?

L133-145 (RNA isolation…): How many technical replicates for the qPCR (I assume RT-qPCR)? Please provide a RefSeq accession number for the CsUbiquitin. The use of a single gene as internal reference should be backed up with evidence that this gene is stable within the assessed conditions/samples. Was a geNorm analysis performed? There is a mention of qPCR using Bio-Rad CFX and RT-qPCR using Applied Biosystems 7500? Please clarify.

L136: 2g RNA? This sounds like a lot. Initial amount of tissues was 100 mg.

Validity of the findings

The impact of this study is not really emphasized. I understand that cannabis/hemp has potential for various uses of this fiber, and I understand the roles of CesA and Csl genes with regards to cellulose/hemicellulose synthesis, but perhaps examples of how those genes were manipulated (or some variants selected) in other species to achieve stronger fiber, or more abundant fiber, could have helped.

The validity of the findings are hard to judge, as many aspects of the methods are missing or unclear (see above). Duplication events are mentioned, but it is not clear how they were identified as such. Exon-intron structures are reported, but since they were not manually curated, they could simply be the results of mis-predictions. Absence of bootstrap support on the phylogenetic tree and absence of statistical analyses on the RT-qPCR data make it impossible to judge the validity of these experiments. It is also unclear what is the usefulness of some of the results presented (e.g., the synteny analysis, the promoter analysis by itself). Moreover, some errors in the naming make it difficult to understand (e.g., CslB is mentioned as being absent in cannabis, but then CslB members are mentioned).

L165: Rephrase as « AtCslB homologs could not been found in cannabis. »

L169: Correct as CsCslE6.

L170: Why only 34 were mapped on the chromosomes? What about the other 20? If the genome used is cs10, the number should be higher than that.

L173: How were duplication events identified?

L183 (and elsewhere): Correct as « promoters »

L174-201: Various sentences are inserted in this section that are not results. For examples « Ethylene accelerates both growth and senescence. ». This does not have its place in this section, and should further supported by references.

L206-L213: Exon-intron structures are reported, but since it is not clear how those genes were obtained, they could be the results of mis-predictions. Did the authors attempt at manual curation to confirm these structures?

L214-L216: I don’t understand the logic of looking at synteny between cannabis and Arabidopsis or rice. Those two species are so far from cannabis that no synteny would be expected anyway. It would make more sense to compare with other species that are closer phylogenetically, and for which synteny (in general) has already been established (Trema, Ziziphus, Morus).

L223-233: As no statistical analyses have been conducted on the RT-qPCR data, it is not clear if any of the tissue-specific expression (or any differences in expression between tissues) is significant. As such, no valid results is presented in this section.

L249: « functions were revealed ». No function were revealed as part of this study. Only assumption based on homology. And without evidence for this homology (no bootstrap displayed on trees), and/or previous demonstration that homologous CesA or Csl proteins share functions, not much can be said.

L251-262: See a previous comment/question on how the naming was achieved. Once this is done, then perhaps an updated tree, or a table with numbers for each subfamily in various plants would be helpful.

L255: Correct « CslB did not exist in cannabis » by « CslB members could not be found in the cannabis genome we investigated ». Other studies on cannabis have shown that cs10 (or CBDRx, often considered the reference cannabis genome) is missing certain genes compared with other, more fragmented, genomes. One example that comes to mind is in the MLO family, where one member (MLO15) was missing in cs10 but not in the other genomes. Again, we don’t know which genome was used here.

L258: Correct as « flax ».

L263-265: The number of CesA genes is claimed to be lower that various plants, but one of the examples has the same number (10) so this sentence is incorrect.

L270-286: Without bootstrap support, it is difficult to judge this section.

L282: The « . » after the parenthesis should be removed.

L288-289: The authors mention that the « number of exons in both CsCslB and CsCslG subfamilies varied from 2 to 9 ». However, in a previous paragraph, they stated that « CslB did not exist in cannabis ». Please clarify.

L296-310: See a previous comment on statistical analyses of RT-qPCR data. Without statistical support, it is impossible to judge this section.

L303-310: The authors discuss a previous study, but it is unclear if the gene names are the same as the naming appears different (CslC04 instead of CslC4). There is also one gene that clearly appears to be an Arabidopsis gene, so I’m not sure why it is mentioned here (AtCesa8 - IRXI). Again, there is a mention of a CslB gene (CslB04), while the authors stated that this family was absent in cannabis (and it is indeed absent on the phylogenetic tree).

Additional comments

Some additional comments on the figures and tables:

Figure 1: Correct as CsCsl. Bootstrap support should be indicated. The legend was incomplete in the version I reviewed, ending by « The tree was divided into seven groups: Group I, G ».

Figure 2: Correct « ekstra cellular » and « cytoplasma » as « extracellular » and « cytoplasm ». Moreover, Cyto_plas is unlikely to means « cytoplasm » as there is another acronym for it: « Cyt ».

Figure 3: Unclear why the title says « 23 WAK/WAKL genes ». Not the good number and not the good gene families. Chromosome numbering in cannabis is usually 1-9 and the X chromosome. Unclear which one is the X chromosome here.

Figure 5: How were the UTRs identified?

Figure 6: Correct as « Syntenic ».

Figure S1: Here, bootstrap support is shown. We can see that CslG is not supported, and thus it is unclear if CsCslG1-3-4-5 are really part of the CslG clade or if they belong to another family. Similarly, without any representative from the CslH clade, it is unclear if the CsCslH members are really part of this family, or perhaps part of the CslB clade.

Table S1: Accession numbers appears to come from different genomes, with some having been obtained from the « official » reference genomes (the XP_ numbers, with their corresponding LOC IDs), while some others appear to have been obtained from other genomes. I’m not certain this is a valid approach to take genes from one genome, and then from a different one. Either use a single genome, or do a proper multiple genome analysis, where you report the different genes from the different genomes. One example of problems that could arise with the approach taken by the authors is with CsClsC5 and CsClsC8. Those 2 genes appear to be closely related on the phylogenetic tree, and comes from different genomes. However, when you look at the gene structures, one appear to be having a very long C-terminal stretch that is absent in the other, which might be driving them to be considered distinct, but could simply be the results of a misprediction. In this context, 2 genes could end up being the same gene.

Table S3: In this table, the chromosomes are numbered 1-9 and X. Have the authors tried to confirm that the non-cs10 genes (those with accession numbers starting with KAF) could be unpredicted in cs10. A simple TBLASTN against the genomic sequence could help confirm this.

---

## Round 0.2 · Major Revisions

While the findings are an improvement over the previous version, several issues hinder the manuscript's overall clarity and scientific impact.

Firstly, the rebuttal letter claims revisions to the phylogenetic tree without any accompanying updates to the methodology section.

The manuscript also suffers from several typographical errors (i.e. line 52), which disrupt the readability and professional quality of the text. These issues persist throughout the manuscript, indicating a need for more rigorous proofreading and editing to meet the publication standards.

Despite the complexity of the data presented, the results section remains unorganized, lacking subdivision into clear, focused subtopics. This presentation could be inpired by the approach seen in Liu et al. (2022) (https://bmcgenomdata.biomedcentral.com/articles/10.1186/s12863-022-01026-0), which has a clear rationale in introduction and results.

Moreover, the manuscript describes maximum likelihood trees but does not include scale bars, which are essential for interpreting the evolutionary distances between sequences. This omission is a fundamental error in phylogenetic analysis, making the results less informative and useful for the reader.

In conclusion, while there are improvements in the data presented compared to previous drafts, the manuscript still requires significant revisions to enhance its scientific rigour, readability, and overall quality. The incorporation of detailed, structured subtopics in the results section, rigorous proofreading, and methodological clarity about updates to phylogenetic analyses are necessary steps before this manuscript can be considered for publication.

Reviewer 2 ·

Basic reporting

The authors should take care of the references format, such as:
Line 52, Line 317, etc. Line 353, where is the reference?

Experimental design

no comment

Validity of the findings

no comment

---

## Round 0.3 · Major Revisions

In this version, parts that need to be better suited for readability and typos were highlighted in yellow. Comments regarding methods were also attached.

**Language Note:** The Academic Editor has identified that the English language must be improved. PeerJ can provide language editing services - please contact us at [email protected] for pricing (be sure to provide your manuscript number and title). Alternatively, you should make your own arrangements to improve the language quality and provide details in your response letter. – PeerJ Staff

---

## Round 0.4 · Major Revisions

I thank the effort and dedication you have put into your revised version. However, I have identified several areas that require improvement before the manuscript can be considered for publication. These revisions are essential to ensure clarity, accuracy, and scientific rigor.

Legibility and Usage of 's
The manuscript contains several instances where legibility is compromised and the usage of 's is not adequate for scientific papers. Scientific texts should maintain clarity and precision to ensure the content is easily understandable. Here are a few examples highlighting these issues:

Line 31-32: "regulated in response to growth development and environmental stimuli." The phrase would be clearer as "regulated in response to growth, development, and environmental stimuli."
Line 36-38: "and were grouped into seven subfamilies (CESA CSLA CSLB CSLC CSLD CSLE and CSLG) according to phylogenetic relationships." This can be improved by adding commas: "and were grouped into seven subfamilies (CESA, CSLA, CSLB, CSLC, CSLD, CSLE, and CSLG) according to phylogenetic relationships."
Line 39-41: "In addition the expressions of 4 CESA and 10 CsCSL genes in flower leaf root and stem organs of cannabis were detected using RT-qPCR." Revised: "In addition, the expression of 4 CESA and 10 CsCSL genes in flower, leaf, root, and stem organs of cannabis was detected using RT-qPCR."
Line 52-54: "Cellulose is synthesized in the plasma membrane and located in the cell wall as microfibrils which is the long chain structure of 14-D-glucose units connected by glycosidic bonds (Mujtaba et al. 2017)." Revised: "Cellulose is synthesized in the plasma membrane and located in the cell wall as microfibrils, which are long-chain structures of β-1,4-D-glucose units connected by glycosidic bonds (Mujtaba et al. 2017)."
Line 67-68: "Cellulose and hemicellulose polysaccharides are synthesized by the enzymes of the cellulose synthase A (CESA) family and cellulose synthase-like (CSL) family respectively." Revised: "Cellulose and hemicellulose polysaccharides are synthesized by the enzymes of the cellulose synthase A (CESA) family and the cellulose synthase-like (CSL) family, respectively."
Line 71-72: "The genes of the cellulose synthase-like (CSL) family grouping in from CSLA to CSLG encode enzymes that generate hemicellulose including xylans xyloglucans mannans glucomannans and β-(1,3;1,4) glucan." Revised: "The genes of the cellulose synthase-like (CSL) family, grouped from CSLA to CSLG, encode enzymes that generate hemicellulose, including xylans, xyloglucans, mannans, glucomannans, and β-(1,3;1,4) glucan."


The authors seem to confuse the concepts of PCR amplification efficiency and specificity. PCR efficiency refers to how well the PCR amplifies the target DNA, typically measured by the rate of product accumulation during the exponential phase. Specificity, on the other hand, pertains to the accuracy of the PCR in amplifying only the intended target sequence without producing non-specific products. For instance, in Line 164, the efficiency and specificity of primers are mentioned together without clear differentiation, which could mislead readers about their distinct roles. The importance of PCR efficiency is exemplified in the recommended literature: https://onlinelibrary.wiley.com/doi/full/10.1002/pei3.10106.

Additionally, the manuscript does not address the calibration condition for qPCR, which is critical for obtaining accurate and reproducible results. It is important to include this information to ensure the reliability of the data.

The manuscript also frequently uses the term "homology" incorrectly when discussing gene relationships. Instead of "homology," the authors should use "orthology" and "paralogy" to describe gene relationships accurately. Orthologous genes are those that diverged after a speciation event and retain the same function, while paralogous genes are those that diverged after a duplication event and may evolve new functions. For example, in Line 274-276, the text should focus on orthology and paralogy rather than homology to accurately describe the relationships.

In the attached document, I have highlighted regions in yellow where I found issues with legibility or other errors. Additionally, I have included comments to provide more specific feedback on those sections.

I appreciate your understanding and cooperation in making these necessary improvements.

---

## Round 0.5 · Minor Revisions

Dear Authors,

I have reviewed your manuscript and would like to suggest some final revisions to improve clarity and accuracy. Firstly, in lines 61 to 65, I recommend reformulating the text to: "In Arabidopsis, ten members of the cellulose synthase (CESA) family have been identified, designated as CESA1 to CESA10. Among these, CESA1, CESA3, and one member from the group consisting of CESA2, CESA5, CESA6, or CESA9, are assembled into the Cellulose Synthase Complex responsible for cellulose synthesis in the primary cell wall (Zhang et al., 2021). Another distinct Cellulose Synthase Complex, composed of CESA4, CESA7, and CESA8, is involved in cellulose synthesis in the secondary cell wall (Taylor et al., 2003)." Additionally, please verify the typo "glucomannas" in line 68, as it seems it should be "glucomannans". For line 98, check if the reference should be Douchkov et al.

Overall, the introduction does not clearly state what the motivation of the study is. Consider strongly reformulating lines 100-102 to include a clear statement of the research motivation.

In lines 185-186 (methods), it is unclear why the authors removed the melting curve analysis. If the melting curve analysis was in fact done, please do not remove this information.

The 2-ΔΔCT method mentioned in line 186 requires a calibrator, a treatment called control. Please indicate this in the methodology (see: https://www.gene-quantification.de/abi-3.gif).

Finally, in Figure 7 legend, there is a typo "diffrenet" which should be corrected to "different". Additionally, instead of using letters a-n, refer to each subplot by the gene name (CsCSLA, CsCSLB, etc.) for better clarity.


I hope these suggestions are helpful.

---

## Round 0.6 · accepted · Accept

This manuscript is ready for publication.

Reviewer 2 ·

Basic reporting

no comment

Experimental design

no comment

Validity of the findings

no comment

Additional comments

no comment